# Gel-Based Proteomic Identification of Suprabasin as a Potential New Candidate Biomarker in Endometrial Cancer

**DOI:** 10.3390/ijms23042076

**Published:** 2022-02-14

**Authors:** Fulvio Celsi, Lorenzo Monasta, Giorgio Arrigoni, Ilaria Battisti, Danilo Licastro, Michelangelo Aloisio, Giovanni Di Lorenzo, Federico Romano, Giuseppe Ricci, Blendi Ura

**Affiliations:** 1Institute for Maternal and Child Health—IRCCS Burlo Garofolo, 65/1 Via dell’Istria, 34137 Trieste, Italy; fulvio.celsi@burlo.trieste.it (F.C.); lorenzo.monasta@burlo.trieste.it (L.M.); michelangelo.aloisio@burlo.trieste.it (M.A.); giovanni.dilorenzo@burlo.trieste.it (G.D.L.); federico.romano@burlo.trieste.it (F.R.); giuseppe.ricci@burlo.trieste.it (G.R.); 2Department of Biomedical Sciences, University of Padova, 35131 Padova, Italy; ilaria.battisti@studenti.unipd.it; 3Proteomics Centre, University of Padova and Azienda Ospedaliera di Padova, 35131 Padova, Italy; 4CRIBI Biotechnology Center, University of Padova, 35131 Padova, Italy; 5ARGO Laboratorio Genomica ed Epigenomica, AREA Science Park, Basovizza, 34149 Trieste, Italy; danilo.licastro@cbm.fvg.it; 6Department of Medicine, Surgery and Health Sciences, University of Trieste, 34129 Trieste, Italy

**Keywords:** endometrial cancer, mass spectrometry, serum proteome, Suprabasin, 2D-DIGE, Western blotting

## Abstract

Endometrial cancer (EC) is the most frequent gynaecologic cancer in postmenopausal women. We used 2D-DIGE and mass spectrometry to identify candidate biomarkers in endometrial cancer, analysing the serum protein contents of 10 patients versus 10 control subjects. Using gel-based proteomics, we identified 24 candidate biomarkers, considering only spots with a fold change in volume percentage ≥ 1.5 or intensity change ≤ 0.6, which were significantly different between cases and controls (*p* < 0.05). We used Western blotting analysis both in the serum and tissue of 43 patients for data validation. Among the identified proteins, we selected Suprabasin (SBSN), an oncogene previously associated with poor prognosis in different cancers. SBSN principal isoforms were subjected to Western blotting analysis in serum and surgery-excised tissue: both isoforms were downregulated in the tissue. However, in serum, isoform 1 was upregulated, while isoform 2 was downregulated. Data-mining on the TCGA and GTEx projects, using the GEPIA2.0 interface, indicated a diminished SBSN expression in the Uterine Corpus Endometrial Cancer (UCEC) database compared to normal tissue, confirming proteomic results. These results suggest that SBSN, specifically isoform 2, in tissue or serum, could be a potential novel biomarker in endometrial cancer.

## 1. Introduction

Endometrial cancer (EC), with an increasing incidence, is the most frequent gynaecologic cancer in postmenopausal women [1]. Most EC cases are in the early stages of the disease [2]. Uterine EC is of two types: type 1 is correlated to oestrogen and comprises 80% of cases, while type 2 is described as an independent oestrogen tumour [3].

Many factors increase the risk of developing EC, such as obesity, age, and type 2 diabetes [4].

At present, no diagnostic test is available for EC screening. Abnormal vaginal bleeding is the most common symptom [5]. Further invasive investigations, such as hysteroscopy [6], are needed to obtain a definitive diagnosis.

A test based on biological fluids can dramatically change the diagnosis and treatment of this disease and contribute to its early detection [7]. In this context, molecular biology techniques are fundamental in the early diagnosis and prediction of a cancer therapy’s benefits [8]. Serum proteomic permits the identification of new biomarkers for the diagnosis and prognosis of EC [9]. Several candidate biomarkers, such as FAM83D [10], ITIH4, CLU, C1R, and SERPINC1, have been previously identified with proteomic technology [11].

Many oncogenes have been identified as possible biomarkers in EC, including EGFR, PI3KCA, K-Ras, HER2/neu, and FGFR2 [12]. In physiological conditions, these genes are inactivated, while their activation would lead to an uncontrollable proliferation of cells [13].

Suprabasin (SBSN) is an oncogene and biomarker in several cancers such as lung carcinoma, salivary adenoid cystic carcinoma, and myelodysplastic syndromes (MDS). The physiological role of SBSN is still unknown [14]. The human SBSN gene is localised in chromosome 19 and consists of five exons and four introns, while mRNA produces three isoforms by alternative splicing [15]. The first two proteoforms of SBSN are well defined [16]: isoform 1 has 590 aa, with a predicted mass of 60.541 Da; while isoform 2, a 247 aa long polypeptide with a predicted mass of 25.335 Da, is a proposed oncogene in human malignancies [16].

2D-DIGE is a modified version of 2D-PAGE, which uses up to four fluorescent tags for protein labelling [11]. This technology was successfully employed for the identification of several biomarkers in cancers [17,18,19] and to characterise new pathways in cancer pathophysiology [20,21].

In this study, for the first time, we quantified the abundance of the two isoforms of SBSN in cancer tissue and serum by using Western blotting and evaluated the possible benefit of this protein as a potential novel biomarker in endometrial cancer.

## 2. Results

### 2.1. Proteomic Study

We used 2D-DIGE coupled with MS for the proteomic study to compare the enriched serum proteomic profile of 10 controls (Cys 3) and 10 EC (Cys 5). Proteomweaver software detected more than 2500 (Figure 1) protein spots in both types of the proteome. After software analysis, 24 protein spots (Table 1) showed a significant alteration (*p* < 0.05) of their volume in EC vs. control samples, with a fold change of ≥1.5 or ≤0.6. Seven of them revealed a fold change ≥1.5 (APOC3, APOC2, APOE, SERPINC1, C1R, SERPINA1, A2M), while 17 proteins indicated a fold change ≤0.6 (APOA1, APOA1, APCS, APOE, CLU, CD5L, CFHR1, VTN, C9, C8A, ALB, C4BPA, IGHM, ITIH2, C1R, SERPINA1, FLG2, SBSN, APOA4, CPS1). Spots of interest were subjected to in-gel digestion and LC-MS/MS analysis, and proteins were identified by searching the MS/MS data against the human section of the UniProt database. All parameters functional in assessing the quality of peptide and protein identifications have been reported in Appendix A.

### 2.2. Western Blotting for SBSN Validation

The altered abundance of 2D-DIGE SBSN in depleted serum was validated by Western blotting. SBSN was chosen for proteomic data validation since it has been previously reported either as an oncogene or as a biomarker in other cancers. For both isoforms, the abundance of SBSN in the enriched serum was validated in 30 controls versus 30 EC patients. The abundance of isoform 2 was lower in EC serum than in controls (*p* = 0.0005 and ROC = 0.7544) (Figure 2), while isoform 1 in serum is not significantly higher than in controls (*p* = 0.2523 and ROC = 0.5867).

The abundance of isoform 1 in tissue was lower in EC than in controls (*p* = 0.0001 and ROC = 0.7928) (Figure 3). Isoform 2 was also lower in EC than in controls (*p* = 0.0001 and ROC = 0.7933).

We calculated the ratio of SBSN-1 between ADK patients and controls, and did the same for SBSN-2. We then calculated the Spearman’s rank correlation between the two ratios, both in the serum and in tissue samples. Figure 4 and Figure 5 show the plotted values of the ratios for serum and tissue samples, respectively. In serum samples, the rank correlation between the two ratios was significant (rho = 0.4433, *p* = 0.0142). In tissue samples, the rank correlation between the two ratios was somehow weaker but still significant (rho = 0.3820, *p* = 0.0372).

### 2.3. Bioinformatic Analysis

We aimed to confirm the proteomic results and investigated the TCGA database to assess SBSN expression in EC compared to normal tissue using the GEPIA2 portal. In Figure 6 we reported SBSN mRNA expression in normal versus EC tissues, showing a slight reduction in gene expression in tumours, although not statistically significant (*p* = 0.072).

For enrichment data, we used g: Profiler classification. This tool categorised the identified proteins into groups according to their molecular function, biological processes, and cellular component (Figure 7). Regarding the molecular function, proteins were categorised into phosphatidylcholine-sterol O-acyltransferase, lipoprotein particle receptor binding, enzyme inhibitor activity, and lipase inhibitor activity, while for biological processes, proteins were classified into complement activation, humoral immune response, high-density lipoprotein particle remodelling, and reverse cholesterol transport. Proteins were organised into blood microparticles, extracellular region, extracellular space, and collagen-containing extracellular matrix for cellular components. Pathway enrichment analysis was performed using the REACTOME tool (Figure 7). Proteins were then grouped into six main pathways: plasma lipoprotein remodelling (APOA4, APOE, APOA1, APOC3, APOC2, ALB), plasma lipoprotein assembly, remodelling, and clearance (APOA4, APOE, APOA1, APOC3, APOC2, ALB, A2M), complement cascade (C4BPA, C1R, CFHR1, C9, APCSVTN, C8A, CLU), post-translational protein phosphorylation (APOE, APOA1, ITIH2, SERPINC1, ALB, SERPINA1), plasma lipoprotein assembly (APOA4, APOE, APOA1, APOC3, APOC2, A2M), and chylomicron assembly (APOA4, APOE, APOA1, APOC3, APOC2).

The top networks in which these proteins were required corresponded to (Figure 8): (1) cell spreading, (2) cellular infiltration, (3) apoptosis, (4) adhesion of immune cells, (5) metastasis, and (6) migration of cells. Four proteins were implicated in cell spreading: A2M, ALB, VTN, and APCS. Seven proteins were implicated in cellular infiltration: ALB, APCS, APO1, APOE, IGHM, SERPINA1, and SERPINC1. Thirteen proteins were involved in apoptosis: APOA1, APOC3, APOE, CD5L, CLU, IGHM, SERPINA1, SERPINC1, VTN, Phosphate, A2M, ALB, and APCS. Nine proteins were involved in the adhesion of immune cells: A2M, APCS, APOA1, APOA4, APOE, CFHR1, CLU, SERPINA1, and VTN. Seven proteins were implicated in metastasis: ALB, APOA1, C1R, CFHR1, CLU, SERPINA1, and SERPINC1. Twelve were involved in the migration of cells: SERPINC1, VTN, A2M, ALB, APCS, APOA1, APOE, CD5L, CFHR1, CLU, IGHM, and SERPINA1.

## 3. Discussion

Biomarkers are crucial to detecting cancers early, improving treatment, and testing the response of ongoing therapies [22] at later stages. Although there has been a great effort in identifying the biomarkers of EC in recent years, none of them have yet reached the clinical stage. Identification of oncoproteins as biomarkers of EC may help develop new diagnostic and therapeutic approaches [23].

This study combined ProteoMiner, 2D-DIGE, and mass spectrometry to identify 24 proteins with different abundances that could act as candidate biomarkers, among which SBSN was chosen for further evaluation due to its potential role as an oncoprotein [14]. The quantification of the two isoforms of SBSN in serum and tissue was performed by Western blotting. These data proved that SBSN was downregulated in EC tissue, where the two isoforms of the protein had a good AUC (area under the ROC curve) (isoform 1 AUC = 0.7928 and isoform 2 AUC = 0.7933). Conversely, SBSN abundance in the serum behaved differently. Isoform 1 in serum had a low AUC (AUC = 0.5867) and did not appear to be a promising biomarker, while isoform 2 reached a better predictive value (AUC = 0.75). The low AUC of isoform 1 and the slight difference in abundance measured for isoform 2 in serum, which affected its reliability as a candidate biomarker, could be related to the release/leakage of the protein from other tissues in addition to the endometrium.

Our data pointed to a higher specificity of both SBSN isoforms in tissues than in serum: again, this could be related to the possibility that other cells or tissues could release SBSN into the bloodstream, thus reducing the differences in abundance observed for these proteins in serum. This occurrence eventually led to a lower sensitivity of SBSN as a putative biomarker when measured in serum compared to its tissue levels.

Further studies are needed to evaluate the performance of SBSN as a biomarker, combined with other serum biomarkers for EC. Taken together, our data in tissue indicated a downregulation of all SBSN isoforms in EC. To further confirm these observations, we performed data mining on the TCGA and GTEx databases, finding that SBSN mRNA expression on EC tissue was lower than in normal uterine tissue; although not statistically significant, this analysis further strengthened our results.

SBSN physiological functions have not yet been entirely ascertained. This protein was originally described as a component of the cornified envelope, which is expressed by corneocytes. Isoform 1 possessed structural features classified as a structural protein, while isoform 2 (and 3) lacked this signature. Post-translational modification has been proposed for both isoforms but demonstrated only for isoform 2. Different shreds of evidence have supported the role of SBSN in the pathogenesis of various kinds of cancer such as ovarian, cervical, and breast carcinomas [24]. In oesophageal squamous cell carcinoma (ESCC), SBSN was proposed as a potential biomarker [25]. In ESCC cell lines, the overexpression of isoform 2 promoted cell growth and proliferation, probably through the WNT/β-catenin signalling pathway [26]. In colorectal cancer, WNT/β-catenin and RAS/ERK signalling pathways interacted with active GSK3β as a mediator [27].

Oncogenes were not necessarily upregulated in carcinogenesis. For example, Cyclin D1 oncogene [28] was downregulated in breast cancer, increasing cell migration. PML, a proto-oncogene, was downregulated in prostate cancer, leading to the downregulation of the cell surface HLA class I molecule and immune escape [29].

Furthermore, in this study, we identified several proteins associated with the adhesion of immune cells. This mechanism played a key role in the recruitment and activation of immune T-cells [30], which are crucial in tumour development.

SERPINA1 was an inhibitor of serine proteases [31]. This protein, in some cases, could act like a tumour-promoting factor, leading to the activation of a variety of oncogenic pathways [32,33]. The upregulation of this protein led to a loss of its immune surveillance function, thus promoting tumour progression [34].

A2M was a plasma protein that acted as an antiprotease, inactivating several proteinases [35]. This activity was associated with cell adhesion modulation, contributing to cancer resistance [36].

Cell spreading was the key mechanism that permitted the cancer cell to invade the other parts of the body [37]. VTN was a cell adhesion and spreading factor found in serum and tissues [38].

T-cell infiltration was associated with a good prognosis in patients in early-stage EC [39]. APOE was a protein associated with lipid particles, carrying lipids between organs via the interstitial fluids and plasma [40]. Pancreatic cancer was characterised by an inflammatory environment that included abundant infiltrating immune cells [41]. In pancreatic cancer, APOE involved the expression of *Cxcl1* and *Cxcl5*, known immunosuppressive factors, leading to immunosuppression [42].

Our data, thus, suggested a possible association of the identified protein with metastatization. IPA analysis correlated the inhibition of metastasis by APOA1, following the literature, which suggested that AIBP, in combination with APOA1, had an anticancer effect on colorectal cancer [43].

SBSN is a secreted protein and, as such, it is very probable that a glycosylated form might be responsible for the apparent high MW observed in the 2-DE map. The fact that the 150 kDa form cannot be detected in our blots may be related to the inability of the commercial antibodies to recognize the protein when heavily glycosylated.

In conclusion, our results quantified the abundance of SBSN in EC, both in tissue and serum. Furthermore, our findings indicated that isoform 2, either in tissue or serum, could be used as a potential novel biomarker in EC. In our opinion, isoform 2 of SBSN should be combined with other biomarkers to reach the validation phase.

## 4. Materials and Methods

### 4.1. Patients

A total of 103 patients (60 non-EC controls and 44 EC patients) were recruited at the Institute for Maternal and Child Health—IRCCS “Burlo Garofolo” (Trieste, Italy)—from 2018 to 2021. All procedures complied with the Declaration of Helsinki and were approved by the Institute’s Technical and Scientific Committee. All patients signed informed consent forms. The median age of patients was 45 years, ranging from 33 to 56 years. As controls, endometrial tissue samples from 30 patients who underwent hysterectomy for symptomatic uterine leiomyomas were obtained. For serum analysis, we used another 30 controls with normal endometrium and whose median age was 42 years, ranging from 32 to 77 years.

The clinical and pathological characteristics of the patients enrolled in this study are described in Appendix A. Controls were chosen by excluding oncologic patients, Human immunodeficiency virus (HIV), Hepatitis B virus (HBV), Hepatitis C virus (HCV) seropositive subjects, and patients with leiomyomas or adenomyosis. EC cases were also selected, ruling out women with other oncologic pathologies, Human immunodeficiency virus (HIV), Hepatitis B virus (HBV), Hepatitis C virus (HCV) seropositive patients, and patients with leiomyomas or adenomyosis.

### 4.2. Serum Sample Collection and Enrichment

To separate serum, blood was centrifugated at 5000× *g* × 5 min. After centrifugation, serum was stored at −80 °C. Serum enrichment of low abundance proteins was achieved using a ProteoMiner column (Bio-Rad Laboratories, Inc., Hercules, CA, USA). In brief, 1 mL of serum was incubated for 2 h at room temperature with ProteoMiner beads. After three cycles of washing with PBS, protein elution was performed from the column with TUC buffer: 7 M urea, 2 M thiourea, 4% CHAPS, and 50 mM Tris pH = 8.5. Subsequently, a second elution was conducted with 4% SDS, 100 mM beta-mercaptoethanol, and the sample was precipitated in methanol and chloroform. The pellets were dissolved in TUC buffer and reunited with the first fraction, and the protein content was determined using the Bradford assay.

### 4.3. Sample Preparation for 2D-DIGE and Gel Image Analysis

For 2D-DIGE analysis, 50 µg of protein of the enriched serum from endometrial cancer patients and controls were labelled with 400 pmol of either Cy5 or Cy3. For internal standards, the samples were pooled and labelled with Cy2. The chemical reaction for protein labelling was carried out by incubating the samples on ice for 30 min in the dark. 1 µL of 10 mM lysine was added to stop the reaction. Following that, proteins were diluted to a final volume of 320 µL in the rehydration buffer: 7 M urea, 2 M thiourea, 2% (*w*/*v*) CHAPS, 65 mM DTT, and 0.24% Bio-Lyte (3–10) (Bio-Rad Laboratories, Inc., Hercules, CA, USA). For 2-DE analysis [44], 4–7 18 cm immobilised pH gradient (IPG) strips (Bio-Rad Laboratories, Inc., Hercules, CA, USA) were rehydrated at 50 V for 12 h at 20 °C, and isoelectric focusing (IEF) was performed in a PROTEAN IEF Cell (Bio-Rad Laboratories, Inc., Hercules, CA, USA) as detailed in [44]. After IEF, IPG strip equilibration was executed with two incubations: the first equilibration in 6 M urea, 2% SDS, 50 mM Tris-HCl (pH 8.8), and 30% glycerol for 5 min, and a second equilibration step performed in 4% iodoacetamide for 10 min. Proteins were separated by SDS-PAGE at a constant voltage of 100 V for 10 h. After electrophoresis, 2-DE gels were scanned with a Molecular Imager PharosFX System (Bio-Rad Laboratories, Inc., Hercules, CA, USA). Molecular weights were determined by Precision Plus Protein Prestained Standards (Bio-Rad Laboratories, Inc., Hercules, CA, USA), covering a molecular weight range from 10 to 250 kDa. Two experimental replicates were performed. Gel analysis was conducted using the MFA (multi fluorescence analysis) module of Proteomweaver 4.0 software (both from Bio-Rad Laboratories, Inc., Hercules, CA, USA) to normalise and quantify protein spots.

### 4.4. Western Blotting

Western blotting was used for SBSN data validation in enriched serum (i.e., treated with ProteoMiner beads, as detailed above) and tissue, as previously described [45]. The control and EC tissues were lysed with 1% NP-40, 50 mM Tris-HCl (pH 8.0), NaCl 150 mM with Phosphatase Inhibitor Cocktail Set II 1× (Millipore, Burlington, VT, USA), 2 mM phenylmethylsulphonyl fluoride (PMSF), and 1 mM benzamidine.

In this study, 30 µg of protein from the tissue and the enriched serum were loaded on a 4–20% precast gel (Bio-Rad) and then transferred to a nitrocellulose membrane. The membrane was blocked with 5% defatted milk in TBS-tween 20 after protein transfer and incubated overnight at 4 °C with 1:1000 diluted primary rabbit polyclonal antibody against SBSN (Abcam). After primary antibody incubation, membranes were washed three times with TBS-Tween 0.05% and incubated with HRP-conjugated anti-rabbit IgG and anti-mouse IgG (1:3000, Sigma-Aldrich; Merck Kagan, Darmstadt, Germany). The protein band signal was visualised using SuperSignal West Pico Chemiluminescent (Thermo Fisher Scientific Inc., Ottawa, ON, Canada). The intensities of the immunostained bands were normalised with the total protein intensities measured by staining the membranes from the same blot with a Red Ponceau solution (Sigma-Aldrich, St. Louis, MO, USA).

### 4.5. Trypsin Digestion and MS Analysis

A preparative 2-DE gel (300 µg of loaded proteins) was run and stained with Coomassie colloidal blue for protein visualisation. After gel decolouration, the spots of interest from 2-DE were digested and analysed by mass spectrometry, as previously described by Ura and colleagues [46]. The spots excised from the gel were washed four times with 50 mM ammonium bicarbonate (AB) and acetonitrile (ACN) (Sigma-Aldrich, St. Louis, MO, USA) and dried under vacuum in a SpeedVac system. For spot digestion, 3 µL of 12.5 ng/µL sequencing grade modified trypsin (Promega, Madison, WI, USA) in 50 mM AB were added. Samples were digested overnight at 37 °C. After digestion, peptide extraction was conducted with three changes in extraction by 50% ACN/0.1% formic acid (FA) (Fluka, Ammerbuch, Germany), and samples were dried under vacuum and stored at –20 °C until mass spectrometry (MS) analysis was performed. Samples were dissolved in 12 µL of 3% ACN/0.1% FA and peptides were separated in a 10 cm pico-frit column (75 μm ID, 15 μm Tip; New Objective) packed in-house with C18 material (Aeris Peptide 3.6 µm XB-C18, Phenomenex) using a nano-HPLC system (Ultimate 3000, Dionex—Thermo Fisher Scientific) coupled with an LTQ-Orbitrap XL mass spectrometer (Thermo Fisher Scientific). H_2_O/FA 0.1% and ACN/FA 0.1% were used as eluents A and B, respectively, and chromatographic separation of peptides were performed at a flow rate of 0.25 μL/min using a linear gradient of eluent B from 3% to 40% in 20 min. A Data Dependent Acquisition (DDA) method was employed: a full scan between 300 and 1700 Da was conducted at high resolution (60,000) on the Orbitrap, and the 10 most intense ions were selected for CID fragmentation and MS/MS data acquisition at low resolution in the linear ion trap. Raw data files were analysed with the software package Proteome Discoverer 1.4 (Thermo Fisher Scientific) interfaced with the Mascot Search Engine (version 2.2.4, Matrix Science, London, UK). MS/MS spectra were searched against the human section of the UniProt database (version September 2020, 75,074 entries) using the following parameters: enzyme specificity was set on trypsin with one missed cleavage allowed; precursor and fragment ion tolerance were 10 ppm and 0.6 Da, respectively. Carbamidomethylcysteine and methionine oxidation were formulated as fixed and variable modifications, respectively. The Percolator algorithm was used to assess the False Discovery Rate (FDR) at the protein and peptide level. Proteins identified with at least three unique peptides with high confidence (FDR < 1%) were considered positive hits.

### 4.6. Bioinformatic Analysis

Gene Expression Profiling Interactive Analysis (GEPIA2) was employed to assess SBSN RNA expression in EC. This tool permits RNA expression analysis from a total of 9736 tumours and 8587 standard samples expunged from the TCGA and GTEx projects. The TCGA-UCEC (The Cancer Genome Atlas Uterine Corpus Endometrial Carcinoma) dataset was explored to assess SBSN expression, and the results were compared with those from the uterus dataset from the GTEx (Genotype-Tissue Expression (GTEx) repository.

Proteins identified by MS were analysed by g: Profiler classification systems and categorised according to their molecular function involvement, biological processes, and protein class. For pathway enrichment, the REACTOME tool was used. We employed the Ingenuity Pathway Analysis (IPA) to generate bio-functions [47]. We considered *p* < 0.01 a statistically significant value in IPA. For the filter summary, we only considered associations where confidence was high (predicted) or that had been observed experimentally.

### 4.7. Statistical Analysis

Differences were considered significant between patients and controls when spots showed a fold change ± 1.5 and satisfied the Mann–Whitney sum rank test (*p* < 0.05). All analyses were conducted with Stata/IC 16.1 for Windows (StataCorp LP, College Station, TX, USA).

## Figures and Tables

**Figure 1 ijms-23-02076-f001:**
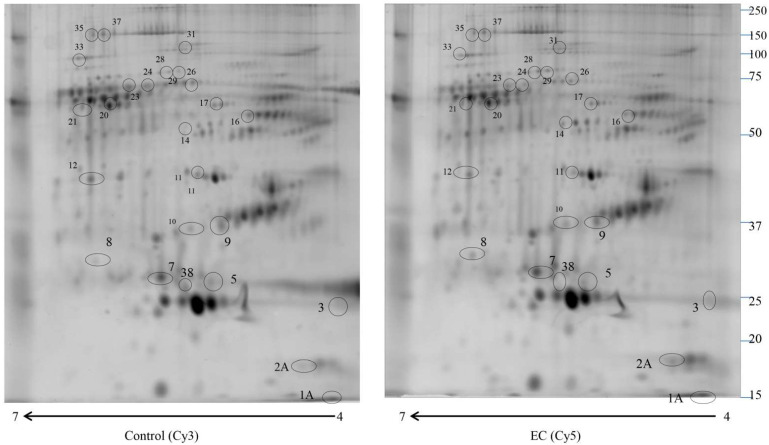
2D-DIGE map of depleted serum from control serum and endometrial serum. IPG strips 4–7 were used for the first dimension, and 10% SDS-PAGE was used for the second dimension. The numbered circles indicate the differently abundant spots.

**Figure 2 ijms-23-02076-f002:**
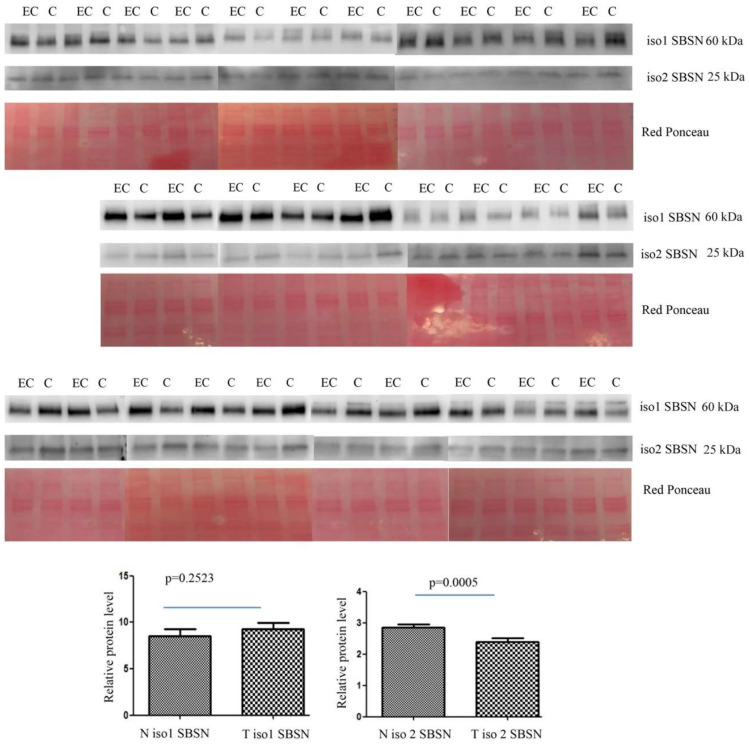
Western blotting analysis of serum isoform 1 and isoform 2 of SBSN in controls (C) and endometrial cancer (EC) patients. The intensity of immunostained bands was normalised against the total protein intensities measured from the same blot stained with Red Ponceau. The graph shows the relative abundance of the two isoforms in control and endometrial cancer serum. Results are shown as a histogram (*p* < 0.05), each bar representing mean ± standard deviation.

**Figure 3 ijms-23-02076-f003:**
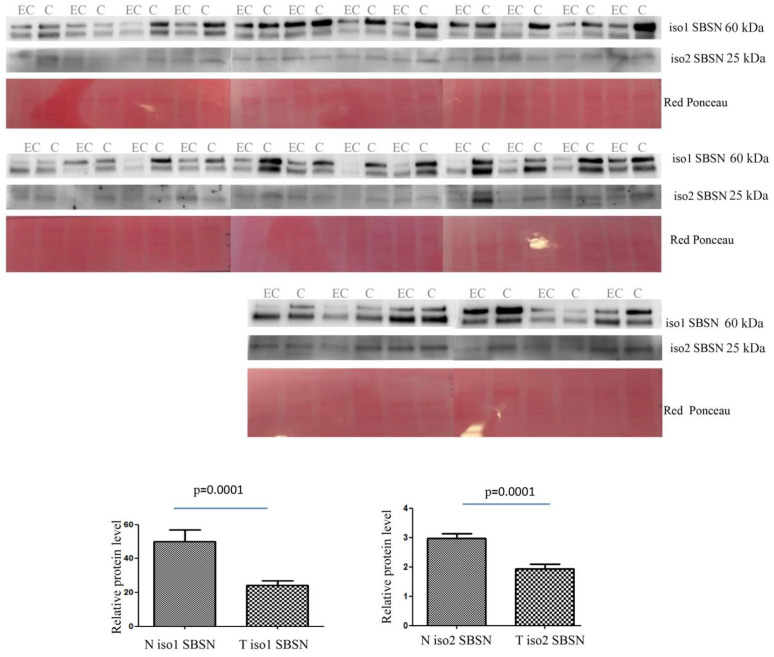
Western blotting analysis of tissue isoform 1 and isoform 2 of SBSN in controls (C) and endometrial cancer (EC) patients. The intensity of immunostained bands was normalised against the total protein intensities measured from the same blot stained with Red Ponceau. The graph shows the relative abundance of the two isoforms in control and endometrial cancer tissue. Results are shown as a histogram (*p* < 0.05), each bar representing mean ± standard deviation.

**Figure 4 ijms-23-02076-f004:**
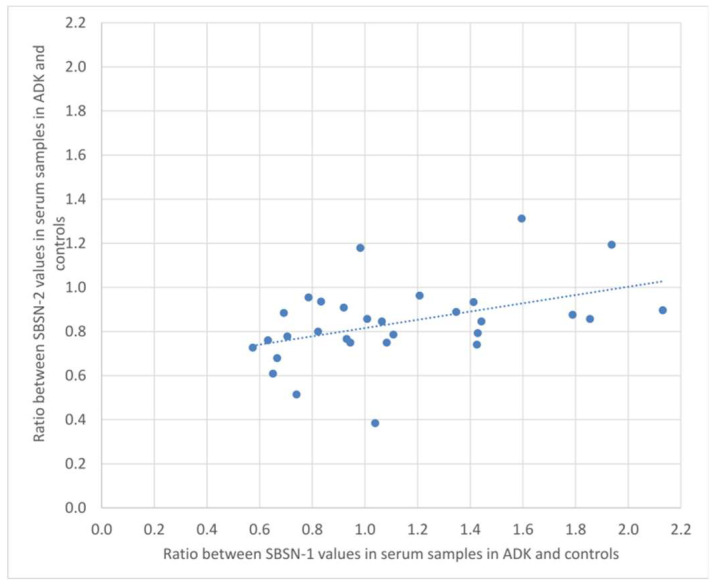
Plotted values of the ratios between SBSN-1 values in ADK vs. controls and the corresponding values for SBSN-2 in serum samples.

**Figure 5 ijms-23-02076-f005:**
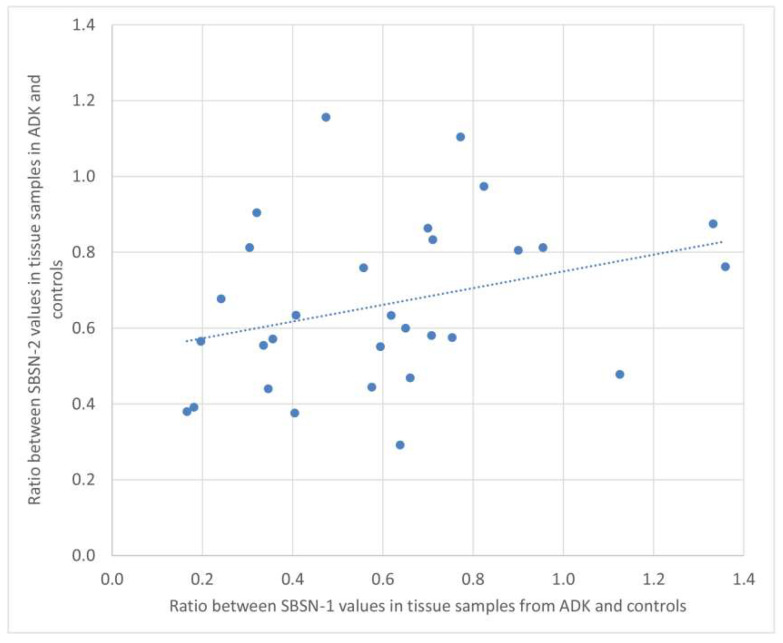
Plotted values of the ratios between SBSN-1 values in ADK vs. controls and the corresponding values for SBSN-2 in tissue samples.

**Figure 6 ijms-23-02076-f006:**
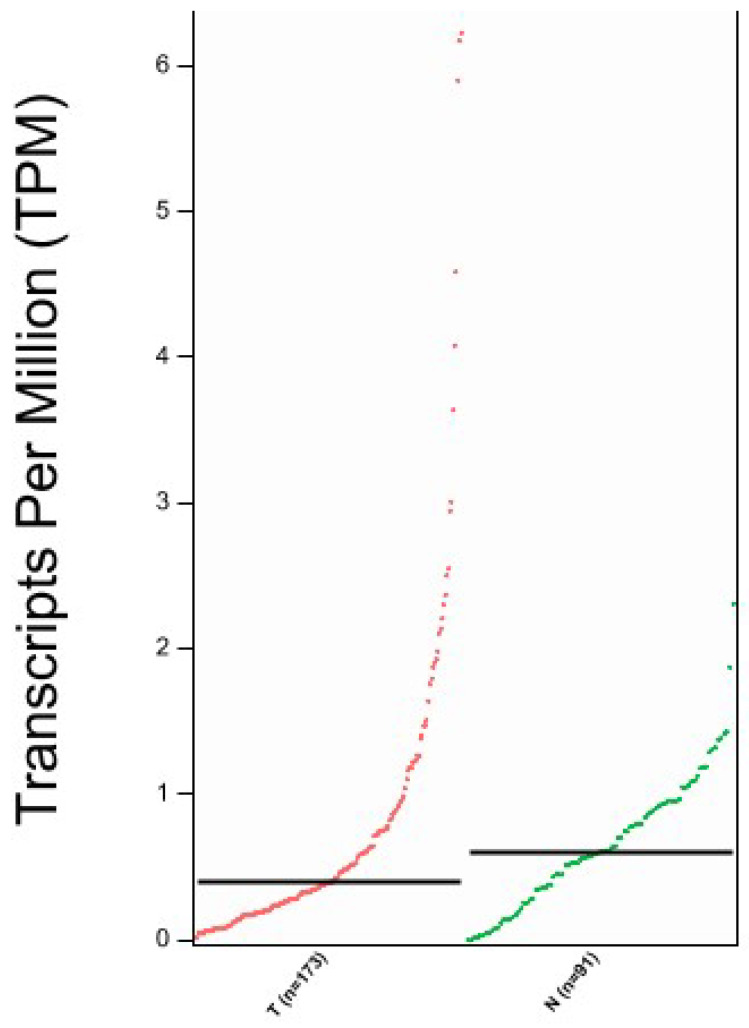
Expression analysis of SBSN mRNA through GEPIA2 interface. T = Tumour tissue, N = Normal Tissue; *n* = number of samples. Data are expressed as number of transcripts of the gene of interest per million of total expressed genes. (TPM) in logarithmic scale. Black transversal line indicates data median.

**Figure 7 ijms-23-02076-f007:**
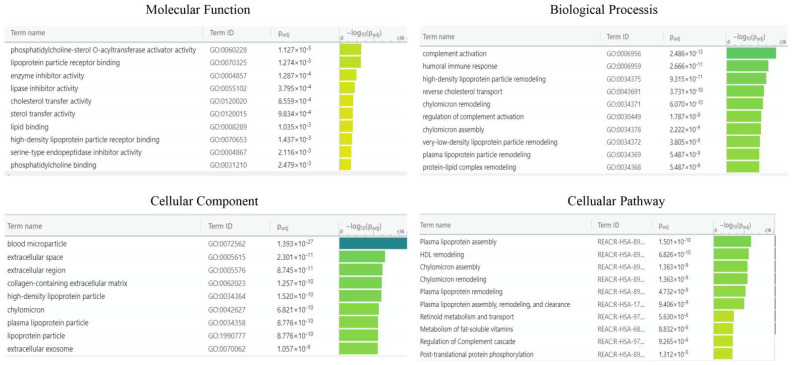
g: Profiler and REACTOME classification of identified proteins in the EC serum according to their molecular function, biological processes, cellular component, and cellular pathway.

**Figure 8 ijms-23-02076-f008:**
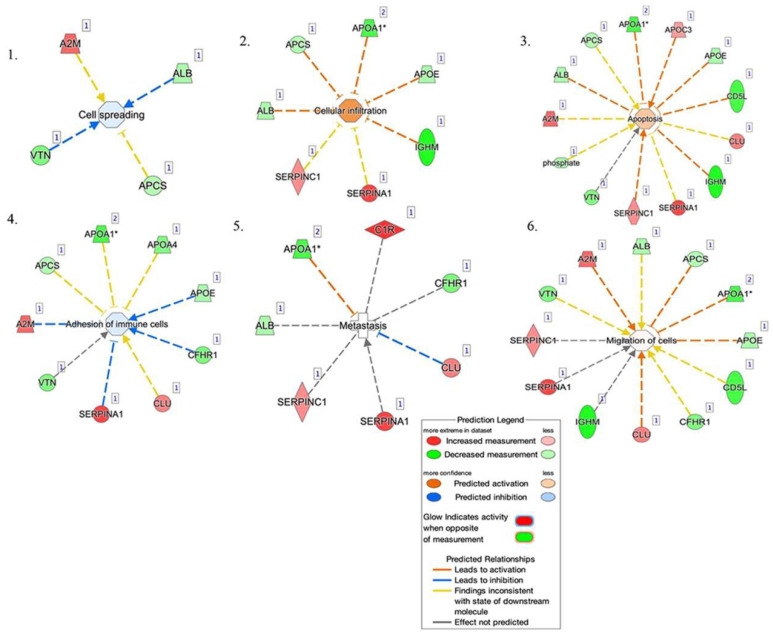
Network build-up from one of the most significant bio-functions: (**1**) cell spreading, (**2**) cellular infiltration, (**3**) apoptosis, (**4**) adhesion of immune cells, (**5**) metastasis, and (**6**) migration of cells.

**Table 1 ijms-23-02076-t001:** Different abundance of proteins identified by mass spectrometry in EC compared to the abundance in control serum.

Accession Number	Spot Number	Protein Description	Gene Symbol	Protein Score	Fold Change *	*p*-Value
A0A3B3ISR2	28	Complement subcomponent C1r	C1R	164.93	4	0.044
P01009	29	Alpha-1-antitrypsin	SERPINA1	325.07	3.66	0.033
P01023	31	Alpha-2-macroglobulin	A2M	150.15	3	0.022
P10909	10	Clusterin	CLU	398.29	2.5	0.033
P01008	14	Antithrombin-III	SERPINC1	403.15	2.22	0.029
P02655	2A	Apolipoprotein C-II	APOC2	152.70	2	0.044
P02656	1A	Apolipoprotein C-III	APOC3	623.84	1.98	0.033
P02743	7	Serum amyloid P-component	APCS	557.00	0.6	0.049
P02649	9	Apolipoprotein E	APOE	324.58	0.6	0.048
P02768	21	Albumin	ALB	1017.31	0.6	0.041
P02748	17	Complement component C9	C9	261.24	0.54	0.021
P07357	20	Complement component C8 alpha chain	C8A	111.76	0.53	0.045
Q5D862	35	Filaggrin 2	FLG2	105.48	0.45	0.036
Q6UWP8	37	Suprabasin	SBSN	156.66	0.43	0.022
P06727	38	Apolipoprotein A-IV	APOA4	844.88	0.4	0.046
P04004	16	Vitronectin	VTN	368.01	0.4	0.021
B1AKG0	12	Complement factor H-related protein 1	CFHR1	354.47	0.39	0.030
P02647	3	Apolipoprotein A-I	APOA1	378.52	0.38	0.028
P31327	33	Carbamoyl-phosphate synthase [ammonia], mitochondrial	CPS1	110.97	0.3	0.036
P02647	5	Apolipoprotein A-I	APOA1	481.70	0.28	0.034
O43866	11	CD5 antigen-like	CD5L	127.16	0.28	0.033
P04003	23	C4b-binding protein alpha chain	C4BPA	373.27	0.24	0.0099

* Fold change is defined as the mean % volume ratio according to the formula: %V = Volume single spot/Volume total spot of EC vs. C.

## Data Availability

Raw data are available upon request to the corresponding author.

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
