# Peer review of "Gel-Based Proteomic Identification of Suprabasin as a Potential New Candidate Biomarker in Endometrial Cancer"

_ijms, 2022, doi:10.3390/ijms23042076_

Round 1

Reviewer 1 Report

None

Author Response

We thanks the reviwer for the revisions

Reviewer 2 Report

Celsi et al. performed comparative proteomic analysis of sera of 10 normal and 10 endometrial cancer suffering individuals by 2D-DIGE analysis. They identified  24 protein spots with significant alteration of their level. They selected one  of them – SBSN – for further analysis in 30 control and EC sera by immunoblotting. They found SBSN isoform 1 was higher whereas SBSN isoform 2 lower in EC sera compared to control sera. Analysis of control and EC tissues showed rather decrease of both SBSN isoforms in EC tissues, which corresponded to data mining GEPIA analysis of tumour tissues. They concluded that the level of SBSN isoform 2 in serum can serve as a potential biomarker of EC.

The findings of the study are potentially interesting, however, the conclusions of the study are not fully supported by the data and there are some issues to be solved before publication. The average differences in SBSN levels are rather minute but with large spread among individuals excluding the potential of SBSN to be used as a biomarker.  

Specific comments

Table 1: SBSN is indicated here as Spot 37. Does it correspond to Spot 37 (MW 150 kDa) on 2-D gel shown in Figure 1? How can be explained low mobility of SBSN-1 (MW 60 kDa)?

Line 100: the statement "while for isoform 1 it was higher in EC than in controls" is not supported by statistics.

Line 109: the statement "These results show that isoform 2 could be a candidate biomarker in serum 109 and tissue." is probably wrong. How can one decide what is sufficiently low level of SBSN-2 to diagnose EC?

The quality of immunoblots (Figure 2 and Figure 3) is not optimal and I doubt the relative quantification to Ponceau S staining is devoid of artefacts (reflecting uneven Ponceau S staining on some lines...). Estimation of signal of some housekeeping gene (beta-actin, gamma-tubulin, GAPDH, SMC1, MCM7, etc.) would better suit as a loading control and more precise tool for semiquantification of SBSN.    

The specificity of Abcam rabbit anti-SBSN antibody should be proved using positive (e.g. skin) and negative controls.

To avoid possible artefacts due to serum depletion by ProteoMiner beads, independent method such as SBSN ELISA should be used for SBSN quantification to confirm the results.

Regarding ratio of SBSN-1 and SBSN-2 isoforms, relative quantitative correlation of both proteins in one sample should be performed and plotted.

Figure 3: The tissue differences of SBSN expression are immense and need to be addressed. Any artefact should be excluded, for instance, differences in location of biopsy, age of donors, etc.

Line 116: Regarding bioinformatic analysis, estimation of mRNA/protein ration would be more informative.   

Minor issues:

Line 80: The sentence "After bioinformatic elaboration, spots of interest were subjected to in-gel 80 digestion, LC-MS/MS analysis, and proteins were identified by searching the MS/MS data 81 against the human section of the UniProt database. All parameters functional to assess the 82 quality of peptide and protein identifications have been reported in File S1 and S2." should be placed more logically in front of preceding sentence starting "Seven of them..."

Figure 2 and 3: Please, show the statistics in graphs. 

Figure 4: The legend should be corrected to correspond to x-axis labelling (EC/T, Ctrl/N).

Please, specify abbreviation AUC.

Author Response

Reviewer 2

Table 1: SBSN is indicated here as Spot 37. Does it correspond to Spot 37 (MW 150 kDa) on 2-D gel shown in Figure 1? How can be explained low mobility of SBSN-1 (MW 60 kDa)?

Our reply: The reviewer’s observation is very pertinent, since indeed, in several cases, the theoretical MW of the identified proteins does not match with the apparent MW, as seen in the 2-DE map. This is not an uncommon phenomenon in gel-based proteomics, since possible isoforms/proteoforms, splice variants, degradation or proteolytic products, possible post-translational modifications, all can contribute to change the apparent pI and MW in 2-DE as compared to the theoretical values derived from the canonical sequences deposited in the protein databases (Chevalier F: Highlights on the capacities of “Gel-based” proteomics. Proteome Sci. 2010, 8: 23. . doi: 10.1186/1477-5956-8-23) (Sameh Magdeldin, Shymaa Enany, Yutaka Yoshida, Bo Xu, Ying Zhang, Zam Zureena, Ilambarthi Lokamani, Eishin Yaoita & Tadashi Yamamoto. Basics and recent advances of two dimensional- polyacrylamide gel electrophoresis. Clin Proteomics. 2014 Apr 15;11(1):16. doi: 10.1186/1559-0275-11-16).

Line 100: the statement “while for isoform 1 it was higher in EC than in controls” is not supported by statistics.

Our reply: Yes, isoform 1 of SBSN in serum is not significantly higher than in controls.

Line 109: the statement “These results show that isoform 2 could be a candidate biomarker in serum 109 and tissue.” is probably wrong. How can one decide what is sufficiently low level of SBSN-2 to diagnose EC?

Our reply: We thank the reviewer for this observation. We want to clarify that our study was not conducted to measure the detection limit of SBSN-2 in serum but to identify putative biomarkers that need to be confirmed on a much more ample cohort of samples and using alternative appoaches. Establishing what would be the level of SBSN-2 below which EC can be suspected or diagnosed, will require a targeted approach for the absolute quantification of the protein. The study described here is based only on the relative quantification of SBSN-2. To take into account the suggestion of the reviewer, we have changed the sentence as follows: “These results indicate that isoform 2 could hold the potential to be a candidate biomarker in serum and tissue”. 

The quality of immunoblots (Figure 2 and Figure 3) is not optimal and I doubt the relative quantification to Ponceau S staining is devoid of artefacts (reflecting uneven Ponceau S staining on some lines...). Estimation of signal of some housekeeping gene (beta-actin, gamma-tubulin, GAPDH, SMC1, MCM7, etc.) would better suit as a loading control and more precise tool for semiquantification of SBSN.

Our reply: To normalize the results of our WB analysis, we determined the total protein content of each sample by Red Ponceau. The reason for this was that, according to our results, the protein that is usually selected as encoded by housekeeping genes (GAPDH) is up-regulated in EC and, thus, not adequate to be used as a control for normalization (Blendi Ura, Lorenzo Monasta, Giorgio Arrigoni, Cinzia Franchin, Oriano Radillo, Isabel Peterlunger, Giuseppe Ricci, Federica Scrimin: A proteomic approach for the identification of biomarkers in endometrial cancer uterine aspirate. Oncotarget. 2017 Dec 12; 8: 109536–109545). In another study of ours the ACTB is down-regulated in the EC serum and not adequate to be used as a control for normalization (Blendi Ura, Stefania Biffi ,Lorenzo Monasta, Giorgio Arrigoni, Ilaria Battisti,Giovanni Di Lorenzo, Federico Romano, Michelangelo Aloisio, Fulvio Celsi, Riccardo Addobbati, Francesco Valle, Enrico Rampazzo ,Marco Brucale, Andrea Ridolfi, Danilo Licastro, Giuseppe Ricci. Two Dimensional-Difference in Gel Electrophoresis (2D-DIGE) Proteomic Approach for the Identification of Biomarkers in Endometrial Cancer Serum. Cancers 2021, 13, 3639). We decided to apply a total protein content normalization method because we could not establish which proteins should be considered as housekeeping in our samples.

The specificity of Abcam rabbit anti-SBSN antibody should be proved using positive (e.g. skin) and negative controls.

Our reply:  The Abcam rabbit anti-SBSN antibody is the only available on the market for western blotting. The specificity of the antibody is used in three human samples by Abcam: serum, Jurkat cell line, urine. Unfortunately, we do not have skin samples for other experiments.

To avoid possible artefacts due to serum depletion by ProteoMiner beads, independent method such as SBSN ELISA should be used for SBSN quantification to confirm the results.

Our reply: The use of ELISA would be very useful for SBSN quantification. Unfortunately, there are no ELISA kits for the two isoforms on the market. Thus, it is not possible to use ELISA for this purpose.

Regarding ratio of SBSN-1 and SBSN-2 isoforms, relative quantitative correlation of both proteins in one sample should be performed and plotted.

Our reply: We added all data in the main text.

Figure 3: The tissue differences of SBSN expression are immense and need to be addressed. Any artefact should be excluded, for instance, differences in location of biopsy, age of donors, etc.

Our reply: : In this study we used 30 endometrial tissue controls (from leiomyoma) and 30 EC tissue samples from the cancer centre. The biopsies were all performed with the same method and always in the body of the uterus. The median age of donors vs EC patients is different. This is because leiomyoma is a pathology occurring during the fertile age, while EC is prevalent in women in menopause. We excluded healthy menopausal patients due to the difficulty in removing the endometrium of menopausal patients which is atrophic and would also have involved removal of the underlying layers of the mucosa. The results of western-blotting in tissue are very clear in both the isoforms, showing that age does not affect the expression of SBSN in the EC. The western-blotting result clearly shows that both the isoforms of EC are downregulated in EC and we feel we can exclude that these results could be due to artifacts.

Line 116: Regarding bioinformatic analysis, estimation of mRNA/protein ration would be more informative. 

Our reply: Evaluation of SBSN expression in normal and carcinoma tissue was performed by database mining, confronting two different datasets. The reviewer’s suggestion, although interesting, is not feasible because neither the TCGA-UCEC dataset nor the GTEx-Uterus dataset contain protein data. Using another dataset for protein data and normalizing it towards RNA data would result in extreme data variability, not providing any meaningful result.

Minor issues:

Line 80: The sentence “After bioinformatic elaboration, spots of interest were subjected to in-gel 80 digestion, LC-MS/MS analysis, and proteins were identified by searching the MS/MS data 81 against the human section of the UniProt database. All parameters functional to assess the 82 quality of peptide and protein identifications have been reported in File S1 and S2.” should be placed more logically in front of preceding sentence starting “Seven of them...”

Our reply: We changed the sentence in: Spot of interest were subjected to in-gel digestion, LC-MS/MS analysis, and proteins were identified by searching the MS/MS data 81 against the human section of the UniProt database. All parameters functional to assess the 82 quality of peptide and protein identifications have been reported in File S1 and S2.”

Figure 2 and 3: Please, show the statistics in graphs.

Our reply: We added the statistics in the graphs.

Figure 4: The legend should be corrected to correspond to x-axis labelling (EC/T, Ctrl/N).

Our reply: We corrected the legend.

Please, specify abbreviation AUC.

Our reply: The abbreviation of AUC has been added and is: Area Under the ROC Curve

Reviewer 3 Report

The manuscript entitled “Gel-based proteomic identification of Suprabasin as a potential new candidate biomarker in endometrial cancer” is an article by Celsi et al. The authors demonstrated that Suprabasin (SBSN) is one of the identified protein that is perturbed when serum/tissue taken from normal patients were compared to endometrial cancer (EC) patients. Taken together, the finding of the manuscripts are of importance yet there are some critical and major issues that need to be addressed which would eventually increase the quality and overall impact of this study.

Specifically :

  • There were several candidates biomarkers (Fold change > 2) such as C1R, SERPINA1, A2M, CLU SERPINC1 were identified by mass spectrometry. Also markers such as APOA1, CFHR1, APOA4, Filaggrin 2 along with selected SBSN were identified (Fold change < 0.5). My question is what make the authors to select just one of it (SBSN) among so many to further analyze it through Bioinformatic analysis. It would have been ideal to pick from the list at least 2-3 from upregulated and 2-3 from downregulated and do the similar analysis. Although some of the candidates have already been described in the literature as pointed out by the authors, yet a comparative analysis necessary to make any decisive point. Also a combination of markers is always an ideal parameter in such type of studies. I would highly recommend to perform this method in order to make this study more concrete.

  • As mentioned, the tissue samples were also procured from EC patients and were used for this study. It would be beneficial to do Immunohistochemistry (IHC) on them to establish the result appropriately.

  • In all the Western Blots, the control protein / loading control is missing for each samples.

Author Response

Reviewer 3

Specifically :

There were several candidates biomarkers (Fold change > 2) such as C1R, SERPINA1, A2M, CLU SERPINC1 were identified by mass spectrometry. Also markers such as APOA1, CFHR1, APOA4, Filaggrin 2 along with selected SBSN were identified (Fold change < 0.5). My question is what make the authors to select just one of it (SBSN) among so many to further analyze it through Bioinformatic analysis. It would have been ideal to pick from the list at least 2-3 from upregulated and 2-3 from downregulated and do the similar analysis. Although some of the candidates have already been described in the literature as pointed out by the authors, yet a comparative analysis necessary to make any decisive point. Also a combination of markers is always an ideal parameter in such type of studies. I would highly recommend to perform this method in order to make this study more concrete.

Our reply:  We thank the reviewer for the question. We selected the SBSN because it is an oncogene previously associated with poor prognosis in different cancers, and a biomarker in several cancers like lung carcinomas, salivary adenoid cystic carcinomas and myelodysplastic syndromes (MDS).  In our previous study we analyzed the association between EC and the putative markers by using a multivariate logistic regression with 4 proteins ( CLU, C1R, SERPINC1, ITIH4) with a very good prediction for EC, with 100% sensitivity and 86.67% specificity with AUC = 0.9289 (Blendi Ura, Stefania Biffi, Lorenzo Monasta, Giorgio Arrigoni, Ilaria Battisti, Giovanni Di Lorenzo, Federico Romano, Michelangelo Aloisio, Fulvio Celsi, Riccardo Addobbati, Francesco Valle, Enrico Rampazzo ,Marco Brucale, Andrea Ridolfi, Danilo Licastro, Giuseppe Ricci. Two Dimensional-Difference in Gel Electrophoresis (2D-DIGE) Proteomic Approach for the Identification of Biomarkers in Endometrial Cancer Serum. Cancers 2021, 13, 3639). In this case we used 30 samples. In our current manuscript the number of samples is larger (103 samples)

and is not possible to validate all this proteins this time. This will be the object of study in future scientific papers.

As mentioned, the tissue samples were also procured from EC patients and were used for this study. It would be beneficial to do Immunohistochemistry (IHC) on them to establish the result appropriately.

Our reply: We have validated the data in both isoforms. The IHC cannot distinguish between the two isoforms. Using IHC would be very useful if we had studied the canonical protein, not the isoform.

In all the Western Blots, the control protein / loading control is missing for each samples.

Our reply: The figures of western blotting are fixed now  

Round 2

Reviewer 2 Report

The authors improved the study, however, there are still some issues left that should be addressed to support the major conclusion of the study. See my 'Reviewer response' to authors comments below. 

Table 1: SBSN is indicated here as Spot 37. Does it correspond to Spot 37 (MW 150 kDa) on 2-D gel shown in Figure 1? How can be explained low mobility of SBSN-1 (MW 60 kDa)?

Our reply: The reviewer’s observation is very pertinent, since indeed, in several cases, the theoretical MW of the identified proteins does not match with the apparent MW, as seen in the 2-DE map. This is not an uncommon phenomenon in gel-based proteomics, since possible isoforms/proteoforms, splice variants, degradation or proteolytic products, possible post-translational modifications, all can contribute to change the apparent pI and MW in 2-DE as compared to the theoretical values derived from the canonical sequences deposited in the protein databases (Chevalier F: Highlights on the capacities of “Gel-based” proteomics. Proteome Sci. 2010, 8: 23. . doi: 10.1186/1477-5956-8-23) (Sameh Magdeldin, Shymaa Enany, Yutaka Yoshida, Bo Xu, Ying Zhang, Zam Zureena, Ilambarthi Lokamani, Eishin Yaoita & Tadashi Yamamoto. Basics and recent advances of two dimensional- polyacrylamide gel electrophoresis. Clin Proteomics. 2014 Apr 15;11(1):16. doi: 10.1186/1559-0275-11-16).

Reviewer response: My question was motivated by findings of several reports showing mobility of SBSN isoforms does not deviate significantly from theoretically estimated MW. The 150 kDa protein was detected by authors as differential by (high quality) 2-D DIGE, but the rest of the study was done on IB signal with unspecified mobility (please, include MW markers on all immunoblots – Figure 2 and 3, etc.). It is still possible that this high 150 kDa molecular form of SBSN is reflecting disease state as well as one of the lower molecular forms. Nevertheless, this 150 kDa signal is not involved in calculations performed for experimental cohorts by IB, thus some (important?) SBSN signal is lost.      

Line 109: the statement “These results show that isoform 2 could be a candidate biomarker in serum 109 and tissue.” is probably wrong. How can one decide what is sufficiently low level of SBSN-2 to diagnose EC?

Our reply: We thank the reviewer for this observation. We want to clarify that our study was not conducted to measure the detection limit of SBSN-2 in serum but to identify putative biomarkers that need to be confirmed on a much more ample cohort of samples and using alternative appoaches. Establishing what would be the level of SBSN-2 below which EC can be suspected or diagnosed, will require a targeted approach for the absolute quantification of the protein. The study described here is based only on the relative quantification of SBSN-2. To take into account the suggestion of the reviewer, we have changed the sentence as follows: “These results indicate that isoform 2 could hold the potential to be a candidate biomarker in serum and tissue”. 

Reviewer response: To better clarify this objection, general understanding "biomarker" is that it should help to distinct ill and healthy individuals, which seems not be the case of SBSN-2 here. It is not about detection threshold of SBSN-2 isoform but about the extent of its decrease in ill individuals to be conclusively 'diagnostic'. My point is though there seems to be statistical difference comparing both cohorts, this difference is too small to be useful in practice for unequivocal decision about the presence of disease on individual basis. Moreover, the levels of SBSN in peripheral blood can be influenced by other (co-)morbid states (see, for instance, Chrastinova, L., et al. (2019). "A New Approach for the Diagnosis of Myelodysplastic Syndrome Subtypes Based on Protein Interaction Analysis." Sci Rep 9(1): 12647.), so I doubt about specific information value of such biomarker. I would recommend to completely remove such statement from the study.  

The quality of immunoblots (Figure 2 and Figure 3) is not optimal and I doubt the relative quantification to Ponceau S staining is devoid of artifacts (reflecting uneven Ponceau S staining on some lines...). Estimation of signal of some housekeeping gene (beta-actin, gamma-tubulin, GAPDH, SMC1, MCM7, etc.) would better suit as a loading control and more precise tool for semiquantification of SBSN.

Our reply: To normalize the results of our WB analysis, we determined the total protein content of each sample by Red Ponceau. The reason for this was that, according to our results, the protein that is usually selected as encoded by housekeeping genes (GAPDH) is up-regulated in EC and, thus, not adequate to be used as a control for normalization (Blendi Ura, Lorenzo Monasta, Giorgio Arrigoni, Cinzia Franchin, Oriano Radillo, Isabel Peterlunger, Giuseppe Ricci, Federica Scrimin: A proteomic approach for the identification of biomarkers in endometrial cancer uterine aspirate. Oncotarget. 2017 Dec 12; 8: 109536–109545). In another study of ours the ACTB is down-regulated in the EC serum and not adequate to be used as a control for normalization (Blendi Ura, Stefania Biffi ,Lorenzo Monasta, Giorgio Arrigoni, Ilaria Battisti,Giovanni Di Lorenzo, Federico Romano, Michelangelo Aloisio, Fulvio Celsi, Riccardo Addobbati, Francesco Valle, Enrico Rampazzo ,Marco Brucale, Andrea Ridolfi, Danilo Licastro, Giuseppe Ricci. Two Dimensional-Difference in Gel Electrophoresis (2D-DIGE) Proteomic Approach for the Identification of Biomarkers in Endometrial Cancer Serum. Cancers 2021, 13, 3639). We decided to apply a total protein content normalization method because we could not establish which proteins should be considered as housekeeping in our samples.

Reviewer response: I agree with authors about reasons to select Ponceau S staining signal for data quantification. However, as the whole study relies solely on this in principle semiquantitative method, it would be more convincing to attempt to support the data by another independent quantitative approach.  

The specificity of Abcam rabbit anti-SBSN antibody should be proved using positive (e.g. skin) and negative controls.

Our reply:  The Abcam rabbit anti-SBSN antibody is the only available on the market for western blotting. The specificity of the antibody is used in three human samples by Abcam: serum, Jurkat cell line, urine. Unfortunately, we do not have skin samples for other experiments.

Reviewer response: Please note, there are several bands with non-stochiometric intensities on the immunoblots shown in supplementary figure. Which one is specific for SBSN? Please, show the whole uncut membrane with detection of both isoforms simultaneously (with two exposition if the SBSN-2 signal is considerably weaker then SBSN-1). Again, the specific signal of SBSN should be proved. There are several anti-SBSN antibodies available (from Abgent, Sigma, etc.) to be tested as an alternative to Abcam antibody. I understand the authors do not have available skin samples, but there should be other ways to perform necessary controls. Cell lines expressing individual SBSN isoforms can be another approach how to test specificity of antibody. To put it in extreme way, it is quite inexpensive to obtain superficial scrapes of human skin (or to collect hair follicles) to prepare control lysates for IB.

Regarding ratio of SBSN-1 and SBSN-2 isoforms, relative quantitative correlation of both proteins in one sample should be performed and plotted.

Our reply: We added all data in the main text.

Reviewer response: The authors added Figures 4 and 5 to plot the SBSN-1 and SBSN-2 ratios in serum and tissues (Figure 4 and 5 look exactly the same). Provided both SBSN-1 and SBSN-2 are detected on the same membrane simultaneously with the same antibody for each individual, the best approach to detect relative changes in levels of both isoforms is to measure their ratio directly (i.e. without any compensation to total protein loading - here to imperfect Ponceau S signal). Such estimation should better support the conclusions of the study.

Author Response

Table 1: SBSN is indicated here as Spot 37. Does it correspond to Spot 37 (MW 150 kDa) on 2-D gel shown in Figure 1? How can be explained low mobility of SBSN-1 (MW 60 kDa)?

Our reply: The reviewer’s observation is very pertinent, since in several cases, the theoretical MW of the identified proteins does not match with the apparent gel mobility, as seen in the 2-DE map. This is not an uncommon phenomenon in gel-based proteomics, since possible isoforms/proteoforms, splice variants, degradation or proteolytic products, possible post-translational modifications, all can contribute to change the apparent pI and MW in 2-DE as compared to the theoretical values derived from the canonical sequences deposited in the protein databases (Chevalier F: Highlights on the capacities of “Gel-based” proteomics. Proteome Sci. 2010, 8: 23. . doi: 10.1186/1477-5956-8-23) (Sameh Magdeldin, Shymaa Enany, Yutaka Yoshida, Bo Xu, Ying Zhang, Zam Zureena, Ilambarthi Lokamani, Eishin Yaoita & Tadashi Yamamoto. Basics and recent advances of two dimensional- polyacrylamide gel electrophoresis. Clin Proteomics. 2014 Apr 15;11(1):16. doi: 10.1186/1559-0275-11-16).

Reviewer response: My question was motivated by findings of several reports showing mobility of SBSN isoforms does not deviate significantly from theoretically estimated MW. The 150 kDa protein was detected by authors as differential by (high quality) 2-D DIGE, but the rest of the study was done on IB signal with unspecified mobility (please, include MW markers on all immunoblots – Figure 2 and 3, etc.). It is still possible that this high 150 kDa molecular form of SBSN is reflecting disease state as well as one of the lower molecular forms. Nevertheless, this 150 kDa signal is not involved in calculations performed for experimental cohorts by IB, thus some (important?) SBSN signal is lost. 

Our reply: This is a very pertinent observation. Following the reviewer’s suggestion we included the MW of markers in all immunoblots. As we mentioned in the first reply the theoretical MW of the identified proteins does not match with the apparent MW, as seen in the 2-DE map. The high MW of SBSN (150 kDa) is most probably due to post-translational modifications. SBSN is a secreted protein and as such it is very probable that a glycosylated form might be responsible for the apparent high MW observed in the 2-DE map. The fact that the 150 kDa form cannot be detected in our blots may be related to the inability of the commercial antibodies to recognize the protein when heavily glycosylated. The only possibility to validate our proteomics data was therefore to use the immunoreactive forms of the protein. This supports therefore the observation of the reviewer regarding the possibility that some important signal related to SBSN might be lost when performing WB. However, considering the data obtained by our 2-DE analysis, it is plausible that if the high MW forms were detected by WB, the validation of the data would be even more consistent. We have now added a sentence in the text to highlight this aspect. 

Line 109: the statement “These results show that isoform 2 could be a candidate biomarker in serum 109 and tissue.” is probably wrong. How can one decide what is sufficiently low level of SBSN-2 to diagnose EC?

Our reply: We thank the reviewer for this observation. We want to clarify that our study was not conducted to measure the detection limit of SBSN-2 in serum but to identify putative biomarkers that need to be confirmed on a much more ample cohort of samples and using alternative approaches. Establishing what would be the level of SBSN-2 below which EC can be suspected or diagnosed, will require a targeted approach for the absolute quantification of the protein. The study described here is based only on the relative quantification of SBSN-2. To take into account the suggestion of the reviewer, we have changed the sentence as follows: “These results indicate that isoform 2 could hold the potential to be a candidate biomarker in serum and tissue”. 

Reviewer response: To better clarify this objection, general understanding "biomarker" is that it should help to distinct ill and healthy individuals, which seems not be the case of SBSN-2 here. It is not about detection threshold of SBSN-2 isoform but about the extent of its decrease in ill individuals to be conclusively 'diagnostic'. My point is though there seems to be statistical difference comparing both cohorts, this difference is too small to be useful in practice for unequivocal decision about the presence of disease on individual basis. Moreover, the levels of SBSN in peripheral blood can be influenced by other (co-)morbid states (see, for instance, Chrastinova, L., et al. (2019). "A New Approach for the Diagnosis of Myelodysplastic Syndrome Subtypes Based on Protein Interaction Analysis." Sci Rep 9(1): 12647.), so I doubt about specific information value of such biomarker. I would recommend to completely remove such statement from the study.  

Our reply: How the reviwer suggests we remove the following sentence from the manuscript: These results indicate that isoform 2 could hold the potential to be a candidate biomarker in serum and tissue.

The quality of immunoblots (Figure 2 and Figure 3) is not optimal and I doubt the relative quantification to Ponceau S staining is devoid of artifacts (reflecting uneven Ponceau S staining on some lines...). Estimation of signal of some housekeeping gene (beta-actin, gamma-tubulin, GAPDH, SMC1, MCM7, etc.) would better suit as a loading control and more precise tool for semiquantification of SBSN.

Our reply: To normalize the results of our WB analysis, we determined the total protein content of each sample by Red Ponceau. The reason for this was that, according to our results, the protein usually for normalization, an housekeeping gene (GAPDH) is up-regulated in EC and, thus, not adequate to be used as a control for normalization (Blendi Ura, Lorenzo Monasta, Giorgio Arrigoni, Cinzia Franchin, Oriano Radillo, Isabel Peterlunger, Giuseppe Ricci, Federica Scrimin: A proteomic approach for the identification of biomarkers in endometrial cancer uterine aspirate. Oncotarget. 2017 Dec 12; 8: 109536–109545). In another study of our group another housekeeping protein  (ACTB) is down-regulated in the EC serum and then not adeguate to be used as a control for normalization (Blendi Ura, Stefania Biffi ,Lorenzo Monasta, Giorgio Arrigoni, Ilaria Battisti,Giovanni Di Lorenzo, Federico Romano, Michelangelo Aloisio, Fulvio Celsi, Riccardo Addobbati, Francesco Valle, Enrico Rampazzo ,Marco Brucale, Andrea Ridolfi, Danilo Licastro, Giuseppe Ricci. Two Dimensional-Difference in Gel Electrophoresis (2D-DIGE) Proteomic Approach for the Identification of Biomarkers in Endometrial Cancer Serum. Cancers 2021, 13, 3639). We decided to apply a total protein content normalization method because we could not establish which proteins should be considered as housekeeping in our samples.

Reviewer response: I agree with authors about reasons to select Ponceau S staining signal for data quantification. However, as the whole study relies solely on this in principle semiquantitative method, it would be more convincing to attempt to support the data by another independent quantitative approach.

Our reply:We agree with the reviewer that a further independent validation method such as a mass-spectrometry based targeted approach for the quantification of SBSN (i.g. SRM) would be beneficial to support our conclusions. However, we do not have access to the required instruments to perform these types of experiments and that’s why we could only rely on the available commercial antibodies to support our conclusions. 

The specificity of Abcam rabbit anti-SBSN antibody should be proved using positive (e.g. skin) and negative controls.

Our reply:  The Abcam rabbit anti-SBSN antibody is the only available on the market for western blotting. The specificity of the antibody is used in three human samples by Abcam: serum, Jurkat cell line, urine. Unfortunately, we do not have skin samples for other experiments.

Reviewer response: Please note, there are several bands with non-stochiometric intensities on the immunoblots shown in supplementary figure. Which one is specific for SBSN? Please, show the whole uncut membrane with detection of both isoforms simultaneously (with two exposition if the SBSN-2 signal is considerably weaker then SBSN-1). Again, the specific signal of SBSN should be proved. There are several anti-SBSN antibodies available (from Abgent, Sigma, etc.) to be tested as an alternative to Abcam antibody. I understand the authors do not have available skin samples, but there should be other ways to perform necessary controls. Cell lines expressing individual SBSN isoforms can be another approach how to test specificity of antibody. To put it in extreme way, it is quite inexpensive to obtain superficial scrapes of human skin (or to collect hair follicles) to prepare control lysates for IB.

Our reply: Following the reviewer’s suggestion we have now provided the uncut membrane with detection of both isoforms simultaneously. Sigma Aldrich and Novus Biological antibodies anti-SBSN are validated only for immunohistochemistry and immunofluorescence while not for western-blotting. Of the four type of antibodies anti-SBSN distributed by ThermoFisher, only two are validated for western blot and they appear to not recognize SBSN-2 (no lower MW bands). Moreover Abgent company apparently has retired this product from the market. The only company that we found that markets this antibody for western blotting is Abcam. According to the reviewer’s observation, we have now used keratinocyte cell line to test this antibody and added this data as supplementary material.

Regarding ratio of SBSN-1 and SBSN-2 isoforms, relative quantitative correlation of both proteins in one sample should be performed and plotted.

Our reply: We added all data in the main text.

Reviewer response: The authors added Figures 4 and 5 to plot the SBSN-1 and SBSN-2 ratios in serum and tissues (Figure 4 and 5 look exactly the same). Provided both SBSN-1 and SBSN-2 are detected on the same membrane simultaneously with the same antibody for each individual, the best approach to detect relative changes in levels of both isoforms is to measure their ratio directly (i.e. without any compensation to total protein loading - here to imperfect Ponceau S signal). Such estimation should better support the conclusions of the study.

Our reply: We detected the western-blotting for both SBSN-1 and SBSN-2 on the same membrane simultaneously with the same antibody for each individual. We confirm the detection of relative changes in levels of both isoforms are made measuring their ratio directly without any compensation to total protein loading.

Reviewer 3 Report

Authors have addressed most of the queries that I raised during my first review report. The manuscript is now improved substantially and seems fit for the publication.

Author Response

We thaks the reviwer for the revision